# Physics of martial arts: Incorporation of angular momentum to model body motion and strikes

**Alexis Merk[1]◎, Andrew Resnick[1,2]◎\***

**1** Department of Physics, Cleveland State University, Cleveland, Ohio, United States of America, **2** Center for Gene Regulation in Health and Disease, Cleveland State University, Cleveland, Ohio, United States of America

◎ These authors contributed equally to this work.
\* a.resnick@csuohio.edu

**Data Availability Statement:** All relevant data are within the manuscript and its Supporting information files.

**Funding:** The authors received no specific funding for this work.

## Abstract

We develop a physics-based kinematic model of martial arts movements incorporating rotation and angular momentum, extending prior analyses. Here, our approach is designed for a classroom environment; we begin with a warm-up exercise introducing counter-intuitive aspects of rotational motion before proceeding to a set of model collision problems that are applied to martial arts movements. Finally, we develop a deformable solid-body mechanics model of a martial arts practitioner suitable for an intermediate mechanics course. We provide evidence for our improved model based on calculations from biomechanical data obtained from prior reports as well as time-lapse images of several different kicks. In addition to incorporating angular motion, our model explicitly makes reference to friction between foot and ground as an action-reaction pair, showing that this interaction provides the motive force/torque for nearly all martial arts movements. Moment-of-inertia tensors are developed to describe kicking movements and show that kicks aimed high, towards the head, transfer more momentum to the target than kicks aimed lower, e.g. towards the body.

## Introduction

"Martial arts", as considered here, consists of a variety of unarmed combat techniques that use a person's own body to deliver focused strikes against an opponent. While weapons are also incorporated into many forms of martial arts such as Hapkido, we note that our model and analysis can easily be extended to include any non-thrown weapon.

Traditional martial arts (Taekwondo, Karate, Judo, etc.) consist of two classes of movements. One class is typically referred to as 'forms' (Poomsae in Taekwondo, Katas in Karate and Judo), the other class refers to sparring or combat motion. The two classes primarily differ by what happens as the movement nears its end. Contact strikes require 'follow-through' motion similar to other sports, while Poomsae movements terminate abruptly without any obvious follow-through movement. For the remainder of this report, we focus specifically on Taekwondo kicks.

**Competing interests:** The authors have declared that no competing interests exist.

Walker's initial analysis [1] modeled martial arts strikes in terms of 1-D collisions. We incorporate angular motion, resulting in a model with broader applicability: not just striking motions, but Poomsae can also be accounted for. We will also show that our model explicitly requires an action-reaction pair between the person and ground and energy input from muscles.

Prior efforts to model the kinematics of martial arts movement have exclusively focused on linear momentum [2, 3], kinetic energy, or both [1, 4, 5]. Biomechanical models of kicks [6, 7] and punches [8–13] have introduced measurable quantities such as maximum foot/hand velocity, maximum knee velocity, torques and accelerations. We wish to note that a large fraction of the literature is concerned with either injury prevention and safety concerns or with detailed muscle physiology, neither of which are relevant here. In-depth biomechanical analysis of a variety of kicking motions has been reviewed in [14–17]. Kinematic measurements and analysis of specific kicking motions are presented in [18–21]. Collectively, these reports primarily focus on linear motion but also provide some quantitative data regarding execution times and angular ranges of motion from which we can extract limited information about rotation kinematics and dynamics. For example, in [15], during a roundhouse kick the pelvis rotates at a maximum rate of approximately 70 $rad/s$. Similarly, in [14], during the power (downward) stroke of an axe kick the angle between the thighs changes by 173° during 0.35 seconds, resulting in an average angular velocity of almost 80 $rad/s$ during the motion.

Our primary aim of this report is to better model martial arts movements (including board breaking) by explicitly incorporating angular momentum into the analysis. Using video and still images, we will show that the foot and ground form an action-reaction pair used to generate rotational motion, which is then transformed to the striking portion (arm, leg, knee, elbow, foot, fist) by body deformation and finally transferred to the target. For 'Poomsae' motion, the practitioner abruptly stops their motion without contacting any other object, in contradiction to 'follow though' movements in other sports (for example, a baseball pitcher's throwing motion or batter's swinging motion).

## Materials and methods

The individual shown in this manuscript has given written informed consent (as outlined in PLoS consent form) to publish these case details.

### Linear and angular momentum for rectilinear motion

Introducing angular momentum into our analysis provides an important pedagogical opportunity to compare and contrast linear and angular momenta. Students may know that total momentum usually refers to total **linear** momentum and that angular momentum is a different "flavor" of momentum, but there is rarely a discussion (beyond unit analysis) why these two kinds of momentum are considered distinct in Physics. Thus, we provide the following example relating linear and angular motion as a useful pedagogical tool to introduce some non-intuitive aspects of rotating frames of reference often passed over in introductory Physics courses. As we will show, these non-intuitive aspects of rotating frames of reference appear when we analyze martial arts movements in terms of collisions, either with targets (strikes) or the ground (initiation of jumping/kicking movements- a collision in reverse).

Consider the following: I am driving a car in a straight line at constant speed "$v$" past a second, stationary, observer who is always at a safe distance from my car. I would say I only possess linear momentum because in my frame of reference relative to the stationary ground, my time-varying position $x(t)$ is given by $x(t) = v * t$. However, the stationary observer would

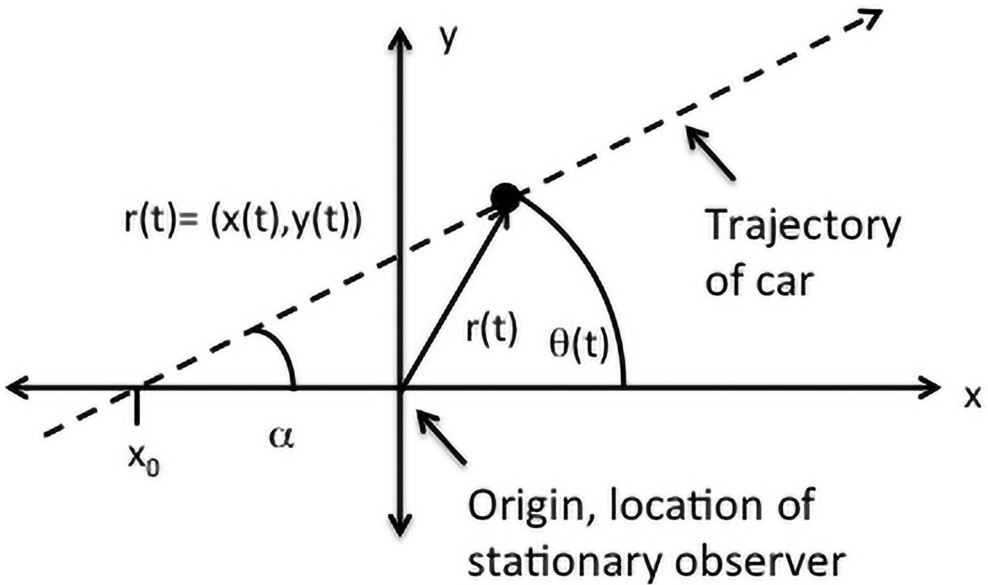

**Fig 1. Equation of a line in polar coordinates.** Equation of a line in polar coordinates.

parameterize my position in terms of a separation distance 'r' and angle $\theta$, both changing in time: r(t) and $\theta(t)$, see Fig 1.

The slope "m" of the line $y = m * x + b$ is given by $m = \tan(\alpha)$ and the y-intercept $b = -x_0 \tan(\alpha)$.

The equation of this line in polar co-ordinates is found using the relationship $x(t) = r(t) \cos \theta(t)$ and $y(t) = r(t) \sin \theta(t)$ to yield:

$$r(t) = \frac{-x_0 \tan(\alpha)}{\sin \theta(t) - \tan(\alpha) \cos \theta(t)} \tag{1}$$

In terms of velocities, the driver will still report a constant velocity $\mathbf{v} = |v| \cos(\alpha)\mathbf{x} + |v| \sin(\alpha)\mathbf{y}$, while the observer will report a time-dependent velocity $\mathbf{v} = \dot{r}\hat{\mathbf{r}} + r\dot{\theta}\hat{\boldsymbol{\theta}}$. Note, the observer will record both radial and angular velocities. If the stationary observer is always very distant and only records the angular component, for example tracking an airplane flying overhead, they will conclude the driver's velocity is not constant because $r\dot{\theta}$ varies in time.

To resolve the apparent paradox of this example, we note that as motion is recorded by the observer, the observer is rotating and thus in a non-inertial frame of reference with respect to the driver. To show this, begin by noting that the driver is in an inertial frame of reference. For example, a glass of water or mug of coffee held by the driver will have a flat liquid surface, identical to the case of no motion. However, the observer reports a time-varying velocity, requiring an applied force. This apparent ("ficticious") force is due to the observer being in a non-inertial frame of reference. Replacing the rotating observer with a direction-sensitive stationary detector does not alter these conclusions as the detector will still record the same time-varying velocity. Finally, claiming the underlying reason for this paradox is that the observer does not have a "full picture" of the motion is faulty as it postulates the existence of an external, "universal", observer- a conceptual device also used to describe the Earth's rotation with respect to a postulated absolute reference frame of the "fixed pattern of stars", and from this derive Coriolis forces for objects moving along the Earth's surface.

By explicitly and carefully relating linear and angular motion, this example serves as a particularly useful way to introduce some counter-intuitive aspects of non-inertial frames of reference, especially the notion of "fictitious forces". This example also serves to show that angular momentum, in contrast to linear momentum, requires a center of rotation, here provided by the observer's location.

## Linear and angular momentum in collision problems

Now we move on to a collision problem incorporating both linear and angular momentum as a model for martial arts movements. While clearly applicable for striking, this model problem can also be applied to Poomase because the (frictional) interaction of the practitioner with the ground is an essential component. For example, jumping off the ground could be modeled as a time-reversed inelastic collision. If the practicioner is initially standing on a movable platform rather than solid ground, the individual roles of linear momentum, angular momentum, and energy can be easily analyzed.

We consider three variations of a person (point mass "m") jumping off a (large) disk with mass "M" (see Fig 2). In the first case, "Case 1", the disk rests on a frictionless surface and the person jumps off the disk in a radial direction. In the second, "Case 2", the disk is attached to the ground by a frictionless pin located through the center of the disk and the person jumps tangentially off the disk edge. In the third case, the disk both rests on a frictionless surface and the person jumps tangentially. In both Case 2 and Case 3, the axis of rotation points out of the page, in the $z$ direction.

These cases can be applied to martial arts movements in a straightforward manner. In terms of strikes, Case 1 could represent a straight-line punch or kick, striking a board or other stationary target and Case 2 would either model (for example) a roundhouse kick or knife-hand punch to a stationary target. Case 3 would model a more complex strike, for example a jumping roundhouse kick where the practitioner is both rotating and airborne at the time of contact. Application of this model to Poomsae is less obvious, but begins with the observation that nearly all martial arts movements, including punches, are initiated and supported by hip rotation to provide power. This rotation requires a frictional interaction between foot and ground to generate torque.

The first two cases are well-known examples often used separately to apply conservation of linear momentum $\mathbf{P}$ to relate final velocities of the person "$\mathbf{v}$" and disk "$\mathbf{V}$" (case 1, $m\mathbf{v} = -M\mathbf{V}$) and angular momentum $\mathbf{L}$ to relate the disk rotation rate with the person's final velocity (Case 2, $I_p\omega = -I_d\Omega \rightarrow mRv = -\frac{1}{2}MR^2\Omega$) where $I_d$ is the moment of inertia of the disk $I_d = \frac{1}{2}MR^2$, $I_p$ the moment of inertia of the person standing on the edge of the disk $I_p = mR^2$, $\omega$ the angular velocity of the person at the time of jumping ($\omega = \frac{v}{R}$), and $\Omega$ the rotation rate of the disk. The third case requires all three conservation laws ($\mathbf{P}$, $\mathbf{L}$ and energy $E$) to hold simultaneously. Note, a worked example related to this case can be found as Example 3.4 in Taylor's "Classical Mechanics" [22]. Analysis of Case 3 proceeds similarly to that example with one significant difference: rather than using force and torque to first determine the impulse initiating motion and then determine the subsequent force- and torque-free motion, we use the conservation laws to directly determine the force- and torque-free motion.

One critical difference between Case 3 and Case 2 is that because the disk is not anchored to the ground, the moment arm for the person is not the disk radius "R" but the distance to the center of mass "l", $l \neq R$. $I_p = ml^2$ and $I_d = \frac{1}{2}MR^2$ are the moments of inertia of the point-mass person and the disk, respectively. Just as the person leaves the disk, we write the final angular momentum as $L_f = mlv + \frac{1}{2}MR^2\Omega$. It is critical to note that once the person has jumped, the motion of both the person and disk each evolve in a force- and torque-free manner; the disc

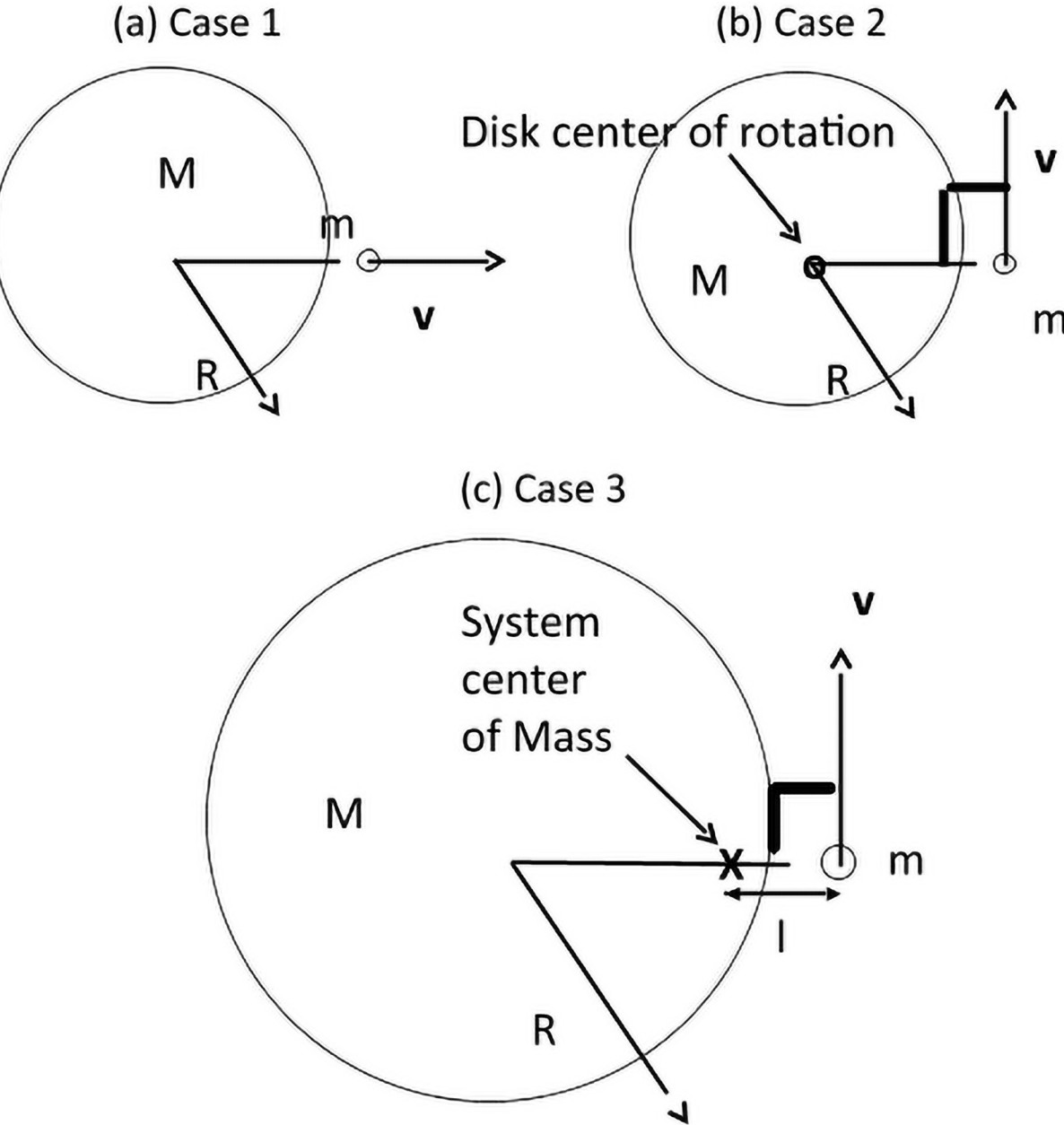

**Fig 2. Three collision examples.** As seen from above, schematic of a person (mass m) standing on the edge of a large platform (mass M). In (a), the person jumps radially from the disk edge at some velocity "v" and (b) jumps tangentially from a disk that is anchored to the ground by a frictionless pivot. In Case (c), the geometry is the same as (b) but the disk is not anchored to the ground by a pivot, the symbol "x" marks the center of mass of disk + person.

will rotate about its center of mass with constant angular velocity, the disk does not rotate about the center of mass of the disk + person. In order for one object to orbit around another, there must be a force acting between the two.

Conservation of energy: $E_{final} = E_{initial} + E_{in} - E_{out}$. Here, we can assume there are no losses to the environment ($E_{out} = 0$), but as we will see, we must retain the term $E_{in}$. Again, $E_{initial} = 0$ and $E_{final} = \frac{1}{2}mv^2 + \frac{1}{2}MV^2 + \frac{1}{2}I_d\Omega^2$.

Because the distance "l" is known, $l = \frac{M}{m+M}R$, these three simultaneous equations with four unknowns "v", "V", "$E_{in}$", and "$\Omega$" are:

$$mv + MV = 0 \tag{2}$$

$$mlv + \frac{1}{2}MR^2\Omega = 0 \tag{3}$$

$$-E_{in} + \frac{1}{2}mv^2 + \frac{1}{2}MV^2 + \frac{1}{2}I_d\Omega^2 = 0 \tag{4}$$

Can be most easily solved for $E_{in}$ and $\Omega$:

$$\Omega = -\frac{2mv}{(m+M)R} \tag{5}$$

$$E_{in} = \left[\frac{m}{2}\left(1 + \frac{m}{M}\right) + \frac{m^2M}{(m+M)^2}\right]v^2 \tag{6}$$

If we had not included $E_{in}$, there would not be a solution to the three conservation equations. Thus, this case usefully demonstrates the need to account for a source of energy: in this case, it is created by muscles as the person jumps. As another example, the source of energy could come from a pre-compressed spring rigidly attached to "M" and allowed to freely extend, imparting kinetic energy to the mass "m".

If we define $M \equiv \epsilon m$, these expressions appear more simply as:

$$\Omega = -\frac{2v}{(1+\epsilon)R} \tag{7}$$

$$E_{in} = \left[\frac{1}{2}\left(1 + \frac{1}{\epsilon}\right) + \frac{\epsilon}{(1+\epsilon)^2}\right]mv^2 \tag{8}$$

These expressions also readily admit two limiting cases: m = M (equal masses) and M>>m (platform much more massive than the point mass). These limiting cases can be illustrative in an instructional setting.

m = M:

$$\Omega = -\frac{v}{R} \tag{9}$$

$$E_{in} = \frac{5}{4}mv^2 \tag{10}$$

M >> m:

$$\Omega = -\frac{2mv}{MR} \rightarrow 0 \tag{11}$$

$$E_{in} = \left[\frac{m}{2} + \frac{m^2}{M}\right]v^2 \rightarrow \frac{1}{2}mv^2 \tag{12}$$

## Results

### Application to Taekwondo movements

Working the above problem "in reverse" models a striking blow: a foot with velocity "v" is incident onto an extended object. In that case, $E_{in}$ corresponds to the energy imparted to the struck object by the foot moving at speed "v". Because the martial artist possess both linear and angular momentum, collisions involve the transfer of both forms of momenta and also energy to the target. We note that the target does not need to rest on a frictionless surface, only that it not be rigidly attached to the ground- for example, by placing the board between two supports.

If we replace the disk by a different shape to better model striking a board or other extended object we simply use the appropriate moment of inertia. For example, the moment of inertia for a thin rod pivoting about an endface is $I\prime = \frac{1}{3}MR^2$. Modeling the striking blow as a collision problem and performing the same prior calculation results in:

$$\Omega = \frac{3mv}{(m + M)R} \tag{13}$$

$$E_{in} = \left[ \frac{m}{2}\left(1 + \frac{m}{M}\right) + \frac{3}{2}\frac{m^2 M}{(m + M)^2} \right] v^2 \tag{14}$$

We can use these equations to answer the following question: Does the mass "m" refer only to the hand or foot or does it also include arm/leg mass, as they also contribute to the initial momentum and energy? By examining previous results [4] showing that at impact, typical striking velocities are approximately between 5–10 m/s and breaking a wooden board requires approximately 5 J of energy. Using test values for the board mass 'M' = 1 kg results in a computed 'm' of approximately 0.2 kg (for v = 5 m/s). Thus, the appropriate mass to use is that of only the hand or foot.

Our results provide insight into the differences between the transfer of energy to the target and transfer of momentum to the target. This problem is typically exemplified in the context of baseball, with Bahill [23] showing that transfer of energy from bat to ball is more important than momentum transfer, meaning use of a lighter bat can result in a longer hit. In the context of martial arts, this problem is phrased in terms of power versus speed of a strike. Our results show that the transfer of power (energy) of a strike to a target is what primarily determines the damage to a target. That is, a fast strike will hit first, but a more powerful strike will result in more damage.

Biomechanically, the motion created by muscles and bones can be modeled in terms of masses and springs, and we are especially interested in rotatory springs (twisting motion). As we will show in images, a roundhouse kick begins by placing a foot securely on the ground and simultaneously with the rear foot kicking motion, the planted front foot rotates so that when the striking foot contacts the target the planted heel points toward the target. During the kicking motion, leg and hip muscles are used to generate torque and rotatory motion with respect to the ground. As this happens, the martial artist will change their moment of inertia by changing their shape in a way analogous to a diver or gymnast. This brings the striking foot into contact at the desired location of the target and also moves their head away from the target in a defensive movement. Combining simultaneous rotation of the hip, rotation of the lower leg, and flexion of the upper body generates a wide variety of kicking motions that are difficult to defensively counter. For example, the martial artist may alter the direction of the rotation axis during a kick so that a low kick may rapidly be brought up to the level of an opponent's head.

Figs 3–5 show three kicking motions, both front and side views: a roundhouse kick, a nadabon kick, and a wheel kick. All three of these show several common elements: bracing a foot

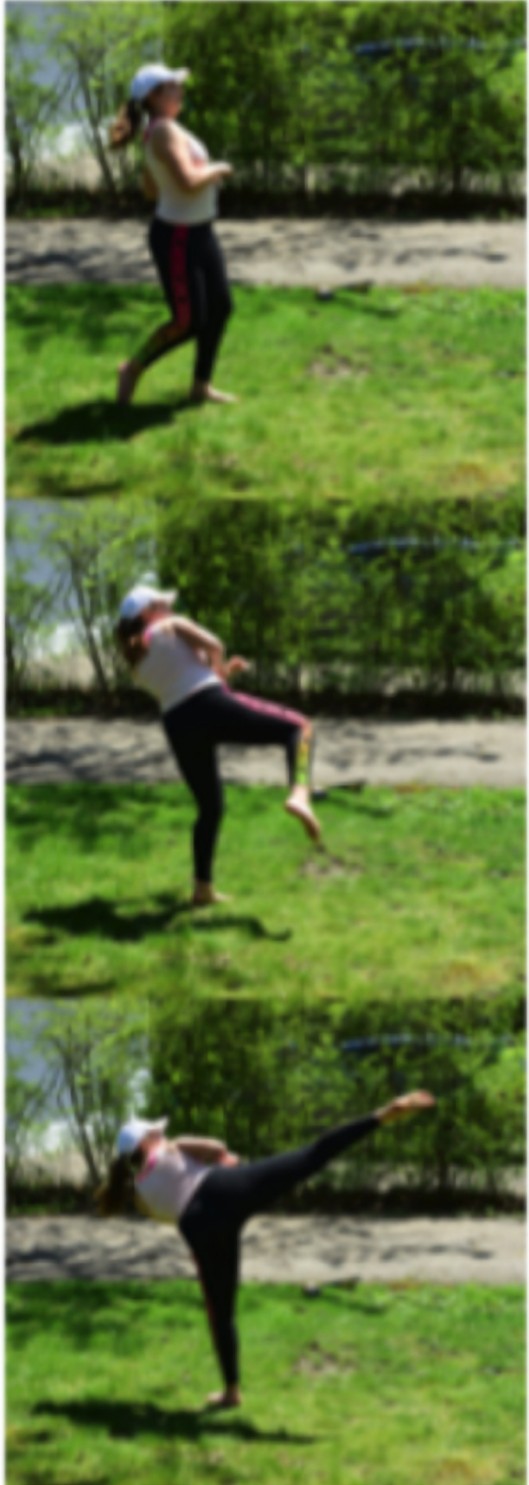
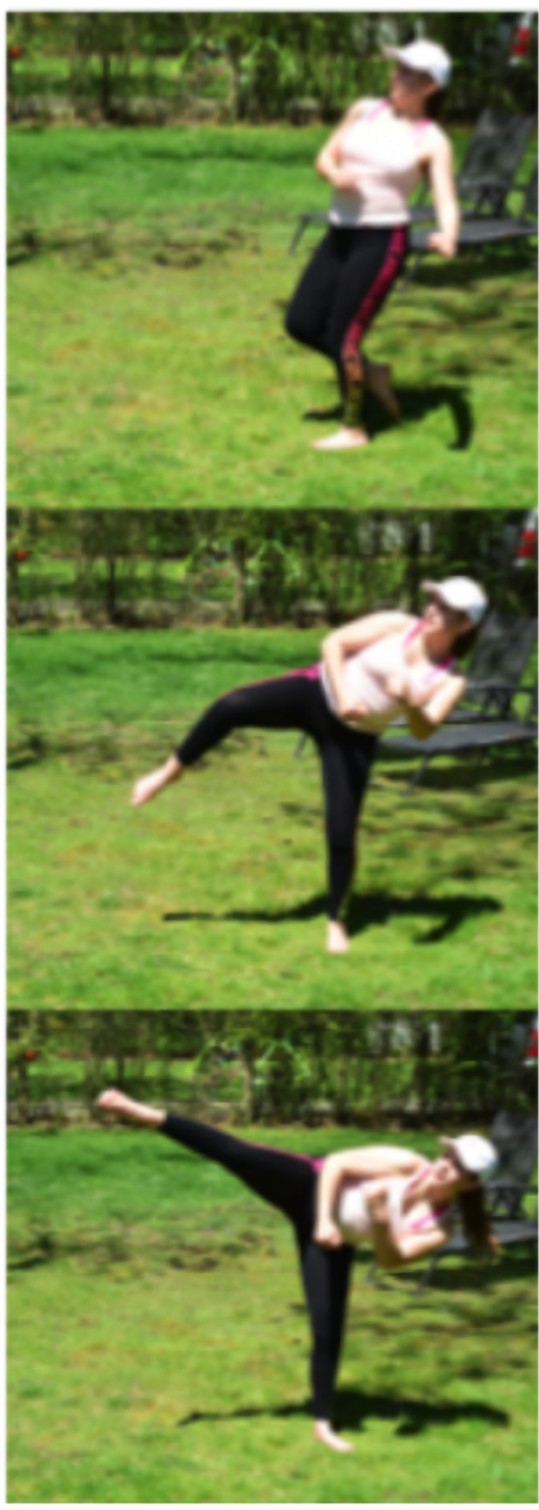

**Fig 3. Roundhouse kick.** Two views of a roundhouse kick, time proceeds downwards. Note the simultaneous rotation of the right front foot, which is used to generate rotation of the hips from frictional contact with the ground, as the rear foot is brought around. Also note the upper body is relatively still in comparison to the lower body. Bending motion of the waist alters the body's rotation axis, bringing the foot up to the opponent's head.

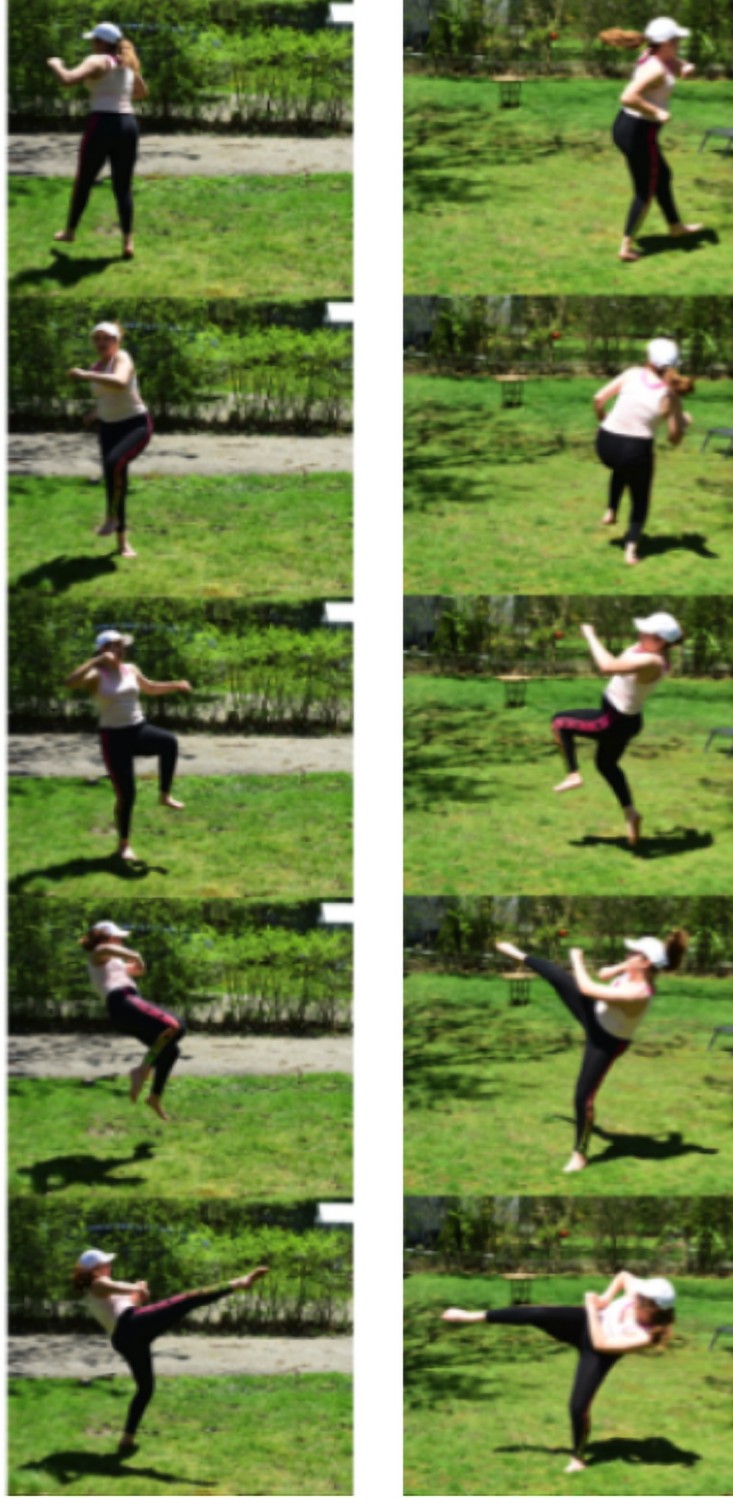

**Fig 4. Nadabon kick.** Two views of a nadabon kick, time proceeds downwards. Here, rotation is initiated by the left rear foot planted on the ground, and rotation of the lower body is accompanied by simultaneous rotation of the upper body. Kicking motion is created by jumping and bringing the rear foot around.

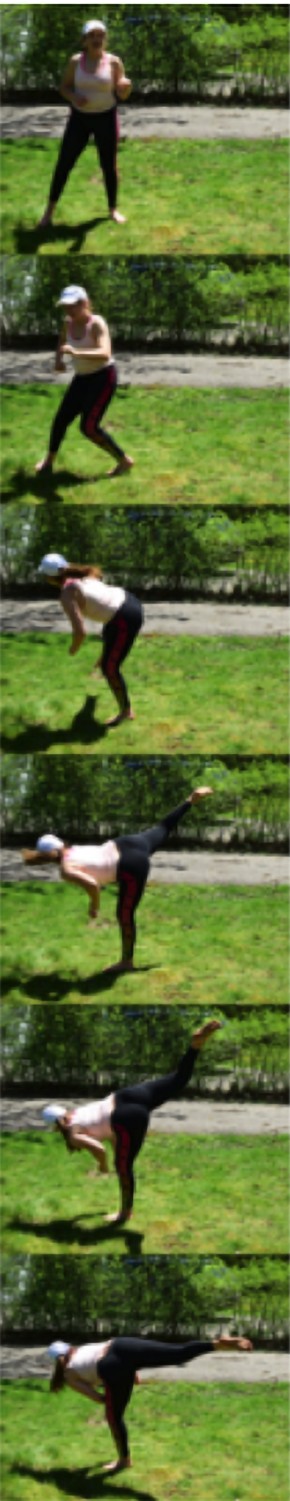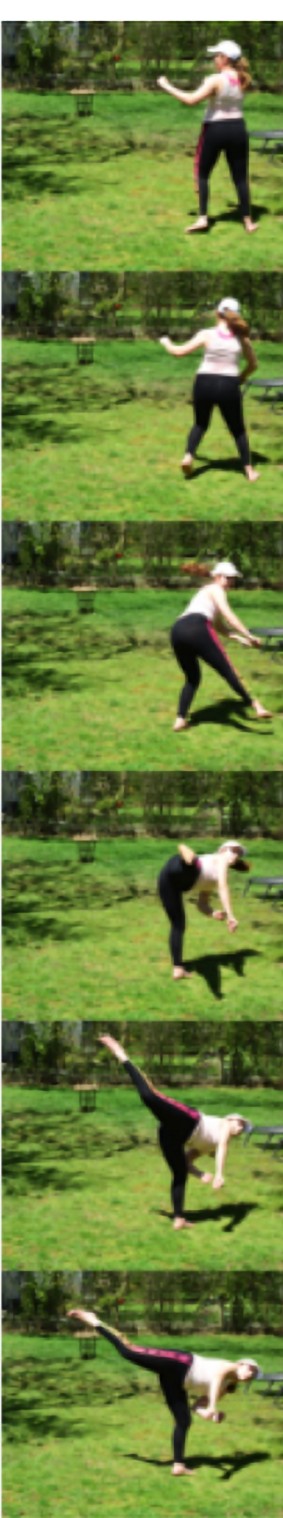

**Fig 5. Wheel kick.** Two views of a wheel kick, time proceeds downwards. Here, rotation is initiated with the right foot, and deformation of the body (bending at the waist) is used to alter the rotation axis, directing the kick at the opponent's head.

against the ground to generate rotational motion, twisting/rotating deformation of the body to maximize rotational movement of a single leg, and flexing at the hips to aim the kicking foot to a specific height at the target. For these image sequences, time progresses when moving down a single column.

Additional videos of these motions are provided as supplemental material with details provided in an Appendix. With these figures as our guide, our next working model moves beyond an Introductory Physics level discussion towards content more appropriate for an intermediate mechanics class, and provides an application of the moment-of-inertia tensor.

## Advanced modeling of the martial artist

We develop here a simple model for a deformable solid body. The essential concept is now "moment of inertia" I. In introductory treatments I is a scalar quantity always proportional to $MR^2$, where R is some representative length (two examples are radius of a disk or distance from a point mass to the axis of rotation). I connects both total angular momentum to angular velocity ($L = I\omega$) and net torque to angular acceleration ($\tau = I\alpha$). Kicking motions alter I, as do grappling motions, but in order to model these motions we must introduce the tensor representation of moment of inertia. The dynamical equations relating total angular momentum to angular velocity and relating net torque to angular acceleration are unchanged, but there are two complications. First, neither angular momentum nor torque are always parallel to angular velocity or angular acceleration. Second, the relevant physical quantities of angular momentum, angular velocity, torque and angular acceleration are no longer "vectors" but rather "pseudovectors" (axial vectors).

We created a simple model to demonstrate these motions (see Fig 6). The junction connecting legs and trunk is considered as a frictionless universal joint, more advanced models could model the junction as a combination of two rotary springs (rotation and elevation).

The moment of inertia tensor relates the resultant motion of a rotating body caused by applied forces and torques. For a cylinder of length L, mass M, and radius 'R' (Fig 7) with coordinate origin located at one endface, the moment of inertia tensor can readily be calculated using the parallel axis theorem:

$$I = \begin{bmatrix} \dfrac{M}{12}(3R^2 + 4L^2) & 0 & 0 \\ 0 & \dfrac{M}{12}(3R^2 + 4L^2) & 0 \\ 0 & 0 & \dfrac{MR^2}{2} \end{bmatrix} \tag{15}$$

$$\equiv \begin{bmatrix} I_1 & 0 & 0 \\ 0 & I_1 & 0 \\ 0 & 0 & I_3 \end{bmatrix}$$

To obtain the moment of inertia tensor for one of the legs, all that is required is to apply a rotation R($\theta$), the transformed tensor is then $I' = R(\theta) * I * R(-\theta)$. An interesting result follows.

Modeling a single leg as a cylinder rotated about the x-axis an angle '$\theta$' gives the result:

$$I' = \begin{bmatrix} I_1 & 0 & 0 \\ 0 & I_1 \cos^2\theta + I_3 \sin^2\theta & (I_1 - I_3)\cos\theta \sin\theta \\ 0 & (I_1 - I_3)\cos\theta\sin\theta & I_1 \sin^2\theta + I_3 \cos^2\theta \end{bmatrix} \tag{16}$$

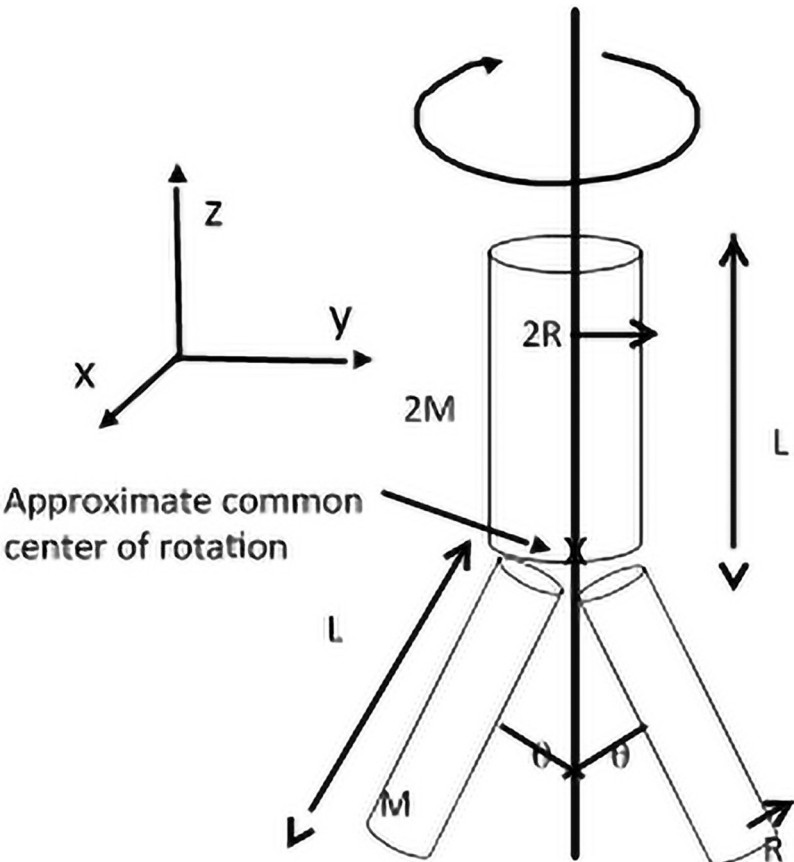

**Fig 6. Abstract model of martial artist.** An abstract model of a martial artist with mass 4M. The trunk is modeled as a cylinder of length 'L', mass '2M' and radius '2R'; each leg is represented by a cylinder of length 'L', mass 'M' and radius 'R' oriented an angle $\theta$ from vertical. Note: the estimated common center of rotation for all three cylinders is marked with an 'x' and is described more fully below.

Examining the off-diagonal components, the moment-of-inertia tensor predicts that rotation of a tilted cylinder (tilt angle less than $\frac{\pi}{2}$ radians) becomes unstable if $L < \sqrt{3}R$, as those off-diagonal components become negative-valued. Interestingly, when both legs are modeled together, splayed open at equal and opposite angles, the combined moment of inertia tensor $I_{legs}$ is found to be:

$$I_{legs} = \begin{bmatrix} 2I_1 & 0 & 0 \\ 0 & 2I_1 \cos^2 \theta + 2I_3 \sin^2 \theta & 0 \\ 0 & 0 & 2I_1 \sin^2 \theta + 2I_3 \cos^2 \theta \end{bmatrix} \quad (17)$$

which is positive-definite, and so is always stable. In practical terms, this means the martial artist may move slowly and even freeze the motion at any time without their body falling over-e.g. a Poomsae movement.

Combining the three cylinders together requires defining a common coordinate system for both legs and trunk, complicated by the geometry of how all three cylinders are joined. To simplify this problem, we imagine the common coordinate origin to be located at the intersection of all three cylinder axes, indicated by the symbol 'x' in Fig 6. This point is approximately

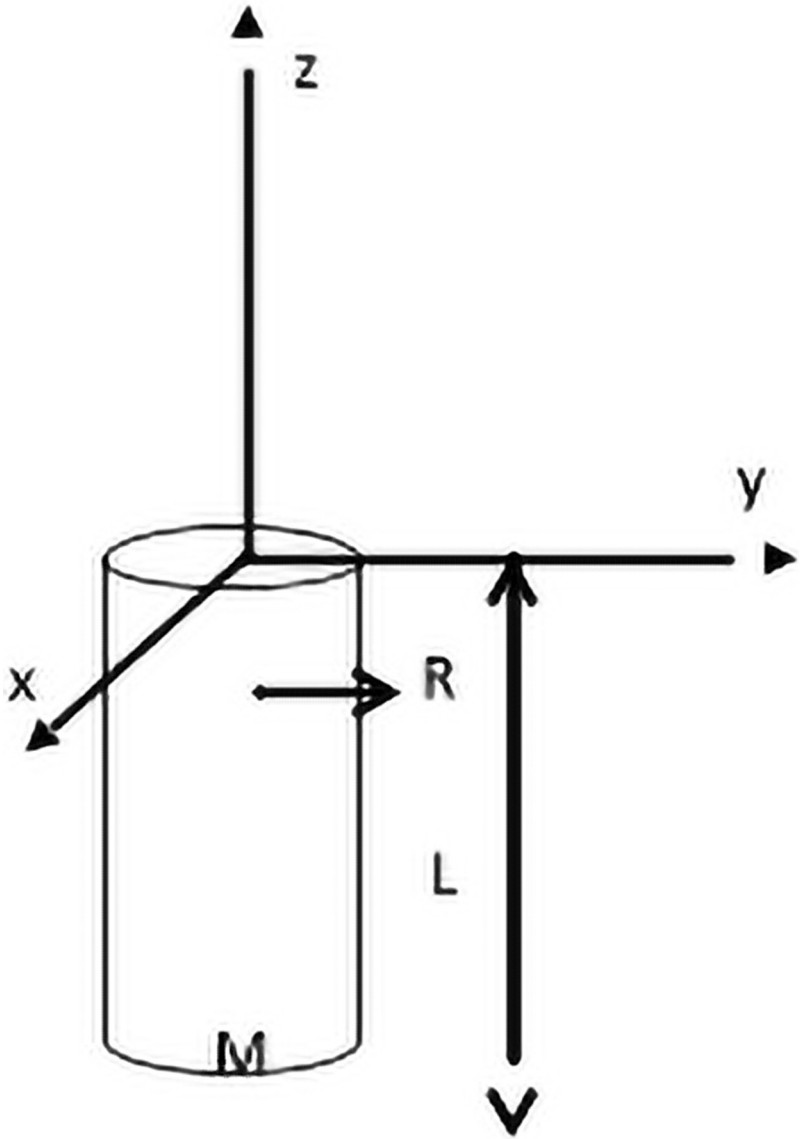

**Fig 7. Moment of inertia of a cylinder.** A cylinder of length L, radius R, and mass M and the coordinates (x,y,z) located with origin at the center of an endface.

located on the trunk cylinder axis a distance 'R' upwards from the bottom endface. Using the parallel axis theorem and noting that a rotation about the $x$-axis of $\pi$ radians does not result in an altered moment of inertia tensor gives:

$$I_{body} = \begin{bmatrix} A & 0 & 0 \\ 0 & B & 0 \\ 0 & 0 & C \end{bmatrix} \tag{18}$$

where A, B, and C are found to be

$$A = M\left(\frac{17R^2}{2} + \frac{3L^2}{2}\right)$$

$$B = M\left[\left(\frac{5R^2}{2} + \frac{2L^2}{3} + RL\right)\cos^2\theta\right] + MR^2\sin^2\theta + M\left(6R^2 + \frac{5L^2}{6} - 2RL\right) \quad (19)$$

$$C = M\left[\left(\frac{5R^2}{2} + \frac{2L^2}{3} + RL\right)\sin^2\theta\right] + MR^2(4 + \cos^2\theta)$$

We again note that the quantities A, B, and C are positive-definite. As an example, analyzing Fig 3, we estimate $\theta = 60°$, M = 15 kg, R = 10 cm and L = 70 cm, resulting in A = $12 kg * m^2$, B = $6.6 kg * m^2$, and C = $5.4 kg * m^2$. If the kick was aimed lower, say at $\theta = 40°$, we find A is unchanged, B is larger (= $8.5 kg * m^2$) and C is smaller (= $3.3 kg * m^2$). In practical terms, by examining the tensor component relating to rotation about the *z*-axis and holding the angular velocity constant, it means the kicker can deliver more momentum and energy to the target by aiming higher because the value of C is larger.

## Conclusion

We have advanced a kinematic model that describes martial arts movements by incorporating, for the first time, angular momentum. We provided pedagogical motivation to distinguish between linear and angular momentum, especially in situations where they appear to interconvert. The first application we provided, appropriate for an introductory Physics course, combines linear and angular angular momentum in a collision problem. The second application is more suitable for an intermediate or advanced mechanics class and introduces moment-of-inertia tensors used to model a deforming object. For both, we provided example calculations demonstrating how the results can be applied to martial arts. Results we obtained for the collision problem can be experimentally tested. For example, the collision results should hold for a martial artist initiating a movement while standing on a freely-rotating platform.

## Supporting information

**S1 Video. Video capture of a roundhouse kick.** Video capture of a roundhouse kick, side-view. Motion is initated by planting the front foot and pushing off the floor with the rear foot. Accompanied by rotation of the waist and hips, the rear foot is brought towards the target. The motion is completed by rotating the front foot and leg so that the heel points towards the target, adding additional rotational momentum and energy to the striking leg. As the striking foot is elevated, the upper body leans back to maintain balance.
(AVI)

**S2 Video. Video capture of a roundhouse kick.** Video capture of a roundhouse kick, front-view. Motion is initated by planting the front foot and pushing off the floor with the rear foot. Accompanied by rotation of the waist and hips, the rear foot is brought towards the target. The motion is completed by rotating the front foot and leg so that the heel points towards the target, adding additional rotational momentum and energy to the striking leg. As the striking foot is elevated, the upper body leans back to maintain balance.
(AVI)

**S3 Video. Video capture of a nadabon kick.** Video Caption: Video capture of a nadabon kick, sideview. The motion begins by planting the rear foot and initiating body rotation while the

front leg rotates an additional amount, planting the front foot with the heel pointing towards the target. As rotation continues, the practitioner hops up, rotating the front foot 180 degrees into kicking position. The practitioner hops again, striking with the left foot as rotational motion continues through the kick. Note how the arms are alternately raised and lowered to provide a small counter rotation used to maintain body control during the kick. (AVI)

**S4 Video. Video capture of a nadabon kick.** Video Caption: Video capture of a nadabon kick, frontview. The motion begins by planting the rear foot and initiating body rotation while the front leg rotates an additional amount, planting the front foot with the heel pointing towards the target. As rotation continues, the practitioner hops up, rotating the front foot 180 degrees into kicking position. The practitioner hops again, striking with the left foot as rotational motion continues through the kick. Note how the arms are alternately raised and lowered to provide a small counter rotation used to maintain body control during the kick. (AVI)

**S5 Video. Video capture of a wheel kick.** Video capture of a wheel kick, sideview. The motion begins by rotating the waist and left leg so that the heel faces the target. Rotational motion continues and as the striking foot is brought to the target, the upper body leans forward to maintain balance. (AVI)

**S6 Video. Video capture of a wheel kick.** Video capture of a wheel kick, frontview. The motion begins by rotating the waist and left leg so that the heel faces the target. Rotational motion continues and as the striking foot is brought to the target, the upper body leans forward to maintain balance. (AVI)

## Author Contributions

**Conceptualization:** Andrew Resnick.

**Investigation:** Alexis Merk.

**Supervision:** Andrew Resnick.

**Writing – original draft:** Alexis Merk, Andrew Resnick.

**Writing – review & editing:** Alexis Merk, Andrew Resnick.

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
