## [Decision Letter · Decision Letter 0]

24 Mar 2021

PONE-D-21-04861

Physics of Martial Arts

PLOS ONE

Dear Dr. Resnick,

Thank you for submitting your manuscript to PLOS ONE. After careful consideration, we feel that it has merit but does not fully meet PLOS ONE’s publication criteria as it currently stands. Therefore, we invite you to submit a revised version of the manuscript that addresses the points raised during the review process.

We look forward to receiving your revised manuscript.

Kind regards,

Mohammadreza Hadizadeh

Academic Editor

PLOS ONE

Journal Requirements:

2. In the manuscript, please clarify how the videos and images were obtained for the study, including how participants were recruited. Please clarify whether any ethical oversight was in place over the video collection, and how the participants gave consent.

3. Please ensure that you refer to Figures 2-7 in your text as, if accepted, production will need this reference to link the reader to each figure.

4. We note that Figures 3, 4 ,5 and supporting information videos include images of participants in the study. 

As per the PLOS ONE policy (http://journals.plos.org/plosone/s/submission-guidelines#loc-human-subjects-research) on papers that include identifying, or potentially identifying, information, the individual(s) or parent(s)/guardian(s) must be informed of the terms of the PLOS open-access (CC-BY) license and provide specific permission for publication of these details under the terms of this license.

Please download the Consent Form for Publication in a PLOS Journal (http://journals.plos.org/plosone/s/file?id=8ce6/plos-consent-form-english.pdf). The signed consent form should not be submitted with the manuscript, but should be securely filed in the individual's case notes.

Please amend the methods section and ethics statement of the manuscript to explicitly state that the patient/participant has provided consent for publication: “The individual in this manuscript has given written informed consent (as outlined in PLOS consent form) to publish these case details”.

If you are unable to obtain consent from the subject of the photograph, you will need to remove the figures / videos and any other textual identifying information or case descriptions for these individuals.

Reviewers' comments:

Reviewer's Responses to Questions

**Comments to the Author**

1. Is the manuscript technically sound, and do the data support the conclusions?

Reviewer #1: Partly

Reviewer #2: Partly

2. Has the statistical analysis been performed appropriately and rigorously? 

Reviewer #1: N/A

Reviewer #2: N/A

3. Have the authors made all data underlying the findings in their manuscript fully available?

Reviewer #1: Yes

Reviewer #2: Yes

4. Is the manuscript presented in an intelligible fashion and written in standard English?

Reviewer #1: Yes

Reviewer #2: Yes

5. Review Comments to the Author

Reviewer #1: In general this is an interesting treatment of martial arts using angular momentum. I like the pedagogical progression looking at both treatments appropriate for an advanced introductory course and an intermediate level mechanics course.

I have one major concern regarding the treatment of Case 3 which should be reviewed by the authors and a second reviewer. I will truthfully admit that angular momentum was never a strength of mine, and a second reviewer should probably be found who teaches intermediate mechanics on a regular basis.

* In Case 3 the person is jumping tangentially from an unconstrained platform. It is not anchored in the center while in Case 2 the platform is anchored with a frictionless axle. In Case 3 the angular momentum for the person is calculated about the center of mass of the system using the length l. On the other hand, there is no correction made for the rotation of the disk itself. Does the disk rotate about its own center of mass or the center of mass of the system? If it rotates about the center of mass of the system, then the parallel axis theorem needs to be applied to the moment of inertia of the platform. If it is rotating about the center of the disk, it is unclear to me why the length l is to be considered the distance to the center of mass. Again, this was never a strength of mine, so it may very well be okay, but it should be triple checked.

* When discussing the equations for this system, l is listed as an unknown, but l can be calculated in terms of m, M, and R. Should this relationship be listed with Eqs 2, 3 and 4?

* In the paragraph starting with line 125 the authors states that the two equations above should hold for a martial artist ... standing on a free-rotating platform. It is not clear which two equations are meant: 7 and 8? 9 and 10? 11 and 12? As well, all of the above equations were derived for Case 3, which had a totally unanchored platform, able to move both translationaly and rotationaly with no fixed axle. Does this really correspond to the situation described in lines 125-127?

* The paragraph at line 149 notes "that the transfer of energy to the target is more significant than the transfer of momentum." While I presume this was meant to follow-up on the previous paragraph, it was not clear to me what constituted "more significant" and why. As well, it is not made clear how we see that the power determines the damage to a target.

* As a tiny detail, E_i and E_f in line 108 should probably be E_initial and E_final for consistency.

Comments regarding figures:

* The description found beginning at line 49 and the Figure 1 itself are a bit confusing. The authors begin by describing the linear situation in terms of one coordinate (x) and discusses a parameterization in terms of theta(t) but makes no mention of r(t). The figure then shows a linear path in two dimensions but does not label the trajectory as the path of the moving object. I fear the figure is a bit busy and unclear. t0 is never mentioned in the text itself, and it can be confusing as to why t0 is not related to x0. The x-intercept is mentioned in the text as b, but the equation for a line in cartesian coordinates shows b as the y-intercept. x tan(theta) should really be the y-intercept in the text, I presume.

I would suggest the authors step back and think about the purpose of the figure. My gut says that the purpose is to show the relation between the cartesian and polar descriptions of the data. I would simply state r(t) = (x(t),y(t)) at the location of interest and label theta(t). It is unclear that the text in the upper right helps any.

From an aesthetic perspective, it would be good if all of the arrowheads are attached to the lines: x-axis, trajectory (though you could probably get away with not worrying about placing an arrow here), and the position vector. The arc describing the angle theta could be more circular with the center roughly on the origin. As well it might help to label the straight line as the path of travel or some such description.

* Figure 2 could probably be fixed up a little as well by placing the tail of the R vector at the visual center of the circle. If the object little m is truly starting at a distance R from the center. It would probably be good to place the object much closer to the rim of the circle.

In general the paper presents an interesting analysis of martial arts using angular momentum at two different levels of complexity. The paper could benefit from an extra pass of copy-editing, some work to clarify and clean up a few of the figures, and a bit of work to make sure that text provides a little extra clarification on a few of the statements. My biggest concern is finding confirmation that Case 3 is properly being modeled regarding the use of the center of mass for the moment arm of the person and inconsistencies that might exist between that and the unconstrained disk.

Reviewer #2: Overall this document is the beginning of some good work, but while the authors developed some interesting ideas, they did not carry them very far or conclude much from them. Furthermore many statements were vague and non-specific. It was poorly organized as it seemed to be three physical situations analyzed without a common thread and scattered about. There was extraneous information that, while true, did not seem pertinent to the analysis. I have some concerns about the accuracy of the development in two places. And finally, there were many editorial errors that need to be cleaned up.

To start, the title “Physics of Martial Arts” is far too broad for what is actually covered in the paper. The title given would more properly be a book title, not a journal article title. I recommend it be altered to be more specific to the exact analyses detailed.

The abstract is vague and non-specific, at least at the beginning. I had to read the entire paper to understand precisely what was accomplished in concrete terms.

I do not see how the entire section “Linear and angular momentum of rectilinear motion” as expressed has anything to do with the rest of the paper. I do not see how Figure 1 is relevant to later sections.

Concerning Case 3 in the section “Linear and angular momentum in a collision problem”, you seem to be making the assumption the disk will rotate about its center of mass when not anchored to the ground as in Case 2. Is that actually true? Wouldn’t it turn about the center of mass of the system instead? Also, if M is much bigger than m the location of the center of mass should be closer to m than presently drawn.

I prime in Equation 4 is not previously defined. I think in the development it’s the same as I sub d. That should be clarified. I understand, however, it’s necessary for the generality of the development when I prime is changed to different values later.

You might overtly state location equation of the center of mass l. It would make the development clearer.

Equation 5 appears to be missing a negative sign.

What is the purpose of the “0.1” in front of “Application to Taekwondo movements”? Things like made the organization of the paper confusing to me. Also, in the “Results” section the inertia tensor of a martial artist is developed. This development should be in the previous section with the results of an analysis of this tensor presented in the “Results” section. I suggest constructing a tight outline for the paper and using that as a guide to its structure.

Concerning line 130, it is not clear to me, and perhaps this is due to my lack of expertise in this specific area, that leaping tangentially off a non-fixed disk platform is the “reverse” of a foot incident on an extended object. At the very least to me there needs to be some kind of change of reference frame. Perhaps the “reverse” of the person leaping off the non-fixed disk platform can be viewed as a foot incident on an extended object in the frame of reference of the extended object. If that’s the case I think that kind of analysis needs to be developed more clearly here.

The analysis of the three kicks was interesting, but I felt the conclusions were weak and non-specific. The motion was described in physical terms, but I’m not sure what was to be concluded from these descriptions.

The moment of intertia tensor model was even better, but the moment of inertia tensor prescribed in Equation 15 is not the moment of inertia tensor of a cylinder in a coordinate system with the origin at the center of an endface as required by Figure 7. It is the moment of inertia tensor in a coordinate system located at the cylinder’s center of mass. Also, Figure 7 did not accurately portray the moment of inertia tensor derived. Either the legs need to be moved towards each other in the figure so the coordinate system originates from the center of the disk face, or the moment of inertia tensor needs to be altered to match the figure. I recommend fixing this and seeing if it changes your conclusions in this section. Furthermore, there was only one conclusion drawn from an analysis of this tensor. I felt like this was incomplete and possibly there were more to be made.

6. PLOS authors have the option to publish the peer review history of their article (what does this mean?). If published, this will include your full peer review and any attached files.

Reviewer #1: No

Reviewer #2: **Yes: **Jeremy C. Holtgrave

---

## [Decision Letter · Decision Letter 1]

8 Jul 2021

PONE-D-21-04861R1

Physics of Martial Arts: Incorporation of angular momentum to model body motion and strikes

PLOS ONE

Dear Dr. Resnick,

Thank you for submitting your manuscript to PLOS ONE. The above manuscript has been reviewed by two of our referees. Comments from the reports appear below.

One of the referees suggests specific minor revisions of your manuscript. When you resubmit your manuscript, please include a summary of the changes made and a brief response to all recommendations contained in the report.

If applicable, we recommend that you deposit your laboratory protocols in protocols.io to enhance the reproducibility of your results. Protocols.io assigns your protocol its own identifier (DOI) so that it can be cited independently in the future. For instructions see: http://journals.plos.org/plosone/s/submission-guidelines#loc-laboratory-protocols . Additionally, PLOS ONE offers an option for publishing peer-reviewed Lab Protocol articles, which describe protocols hosted on protocols.io. Read more information on sharing protocols at https://plos.org/protocols?utm_medium=editorial-email&utm_source=authorletters&utm_campaign=protocols .

We look forward to receiving your revised manuscript.

Kind regards,

Mohammadreza Hadizadeh

Academic Editor

PLOS ONE

Journal Requirements:

Reviewers' comments:

Reviewer's Responses to Questions

**Comments to the Author**

1. If the authors have adequately addressed your comments raised in a previous round of review and you feel that this manuscript is now acceptable for publication, you may indicate that here to bypass the “Comments to the Author” section, enter your conflict of interest statement in the “Confidential to Editor” section, and submit your "Accept" recommendation.

Reviewer #1: (No Response)

Reviewer #2: All comments have been addressed

2. Is the manuscript technically sound, and do the data support the conclusions?

Reviewer #1: (No Response)

Reviewer #2: Yes

3. Has the statistical analysis been performed appropriately and rigorously? 

Reviewer #1: (No Response)

Reviewer #2: N/A

4. Have the authors made all data underlying the findings in their manuscript fully available?

Reviewer #1: (No Response)

Reviewer #2: Yes

5. Is the manuscript presented in an intelligible fashion and written in standard English?

Reviewer #1: (No Response)

Reviewer #2: Yes

6. Review Comments to the Author

Reviewer #1: I appreciate the work that was done in addressing my previous comments. Any concerns that I had have been addressed.

The cleaned-up figures look much better. When looking at Figure 1 and the text, you might want to put x0 back into the figure as that is the only place it is really defined.

In reading back through I found a few small copyediting. I apologize that I did not catch these earlier.

Line 15: "a action-reaction" should be "an action-reaction"

Line 63: I think you mean to say the y-intercept is ... not the x-intercept is (the x-intercept x0)

Line 96: You might want to consider a new sentence with "For example, ..."

Line 108 and beyond: I did not catch the inconsistent use of capitalization with "Case 1", "case 1". My understanding is that any time you refer to cases generically, it would be lowercase (in all three cases), but when you start referring to a specific case (Case 1, Case (c), Case 3) you capitalize it as a proper noun just like you would in an example like "Figure 1 shows ..." versus "the figure shows ..." There are a number of inconsistencies in this region of the paper.

Line 256: a figure reference is missing.

Again, I appreciate the work on the paper, and at this point I have no more comments about the physics or the pedagogy. From my perspective, I recommend the paper for publication once any copyediting issues have been completed to the satisfaction of the Journal Editorial Team.

Reviewer #2: The author's have done exceedingly diligent work addressing all concerns and submitted an outstanding manuscript ready for immediate publication.

7. PLOS authors have the option to publish the peer review history of their article (what does this mean?). If published, this will include your full peer review and any attached files.

Reviewer #1: No

Reviewer #2: **Yes: **Jeremy C. Holtgrave

---

## [Author Response · Author response to Decision Letter 1]

20 Jul 2021

Please see attached file "Authors’ second reply to reviewer comments", thanks.

---

## [Editor Report · Decision Letter 2]

22 Jul 2021

Physics of Martial Arts: Incorporation of angular momentum to model body motion and strikes

PONE-D-21-04861R2

Dear Dr. Resnick,

We’re pleased to inform you that your manuscript has been judged scientifically suitable for publication and will be formally accepted for publication once it meets all outstanding technical requirements.

Kind regards,

Mohammadreza Hadizadeh

Academic Editor

PLOS ONE

---

## [Editor Report · Acceptance letter]

30 Jul 2021

PONE-D-21-04861R2

Physics of Martial Arts: Incorporation of angular momentum to model body motion and strikes

Dear Dr. Resnick:

I'm pleased to inform you that your manuscript has been deemed suitable for publication in PLOS ONE. Congratulations! Your manuscript is now with our production department.

Kind regards,

on behalf of

Dr. Mohammadreza Hadizadeh

Academic Editor

PLOS ONE